# Comprehensive Omics Analysis of a Novel Small-Molecule Inhibitor of Chemoresistant Oncogenic Signatures in Colorectal Cancer Cell with Antitumor Effects

**DOI:** 10.3390/cells10081970

**Published:** 2021-08-03

**Authors:** Tse-Hung Huang, Ntlotlang Mokgautsi, Yan-Jiun Huang, Alexander T. H. Wu, Hsu-Shan Huang

**Affiliations:** 1Department of Traditional Chinese Medicine, Chang Gung Memorial Hospital, Keelung 20401, Taiwan; tsehunghuang_1089@yahoo.com.tw; 2School of Traditional Chinese Medicine, Chang Gung University, Kweishan, Taoyuan 333, Taiwan; 3School of Nursing, National Taipei University of Nursing and Health Sciences, Taipei 112, Taiwan; 4Graduate Institute of Health Industry Technology, Chang Gung University of Science and Technology, Taoyuan 333, Taiwan; 5Research Center for Chinese Herbal Medicine, Chang Gung University of Science and Technology, Taoyuan 333, Taiwan; 6Department & Graduate Institute of Chemical Engineering & Graduate Institute of Biochemical Engineering, Ming Chi University of Technology, New Taipei City 243, Taiwan; 7Ph.D. Program for Cancer Molecular Biology and Drug Discovery, College of Medical Science and Technology, Taipei Medical University and Academia Sinica, Taipei 11031, Taiwan; d621108006@tmu.edu.tw; 8Graduate Institute for Cancer Biology and Drug Discovery, College of Medical Science and Technology, Taipei Medical University, Taipei 11031, Taiwan; 9Division of Colorectal Surgery, Department of Surgery, Taipei Medical University Hospital, Taipei Medical University, Taipei 110, Taiwan; d622101001@tmu.edu.tw; 10Department of Surgery, School of Medicine, College of Medicine, Taipei Medical University, Taipei 110, Taiwan; 11TMU Research Center of Cancer Translational Medicine, Taipei Medical University, Taipei 11031, Taiwan; 12The Ph.D. Program of Translational Medicine, College of Science and Technology, Taipei Medical University, Taipei 11031, Taiwan; 13Clinical Research Center, Taipei Medical University Hospital, Taipei Medical University, Taipei 11031, Taiwan; 14Graduate Institute of Medical Sciences, National Defense Medical Center, Taipei 11490, Taiwan; 15School of Pharmacy, National Defense Medical Center, Taipei 11490, Taiwan; 16Ph.D. Program in Drug Discovery and Development Industry, College of Pharmacy, Taipei Medical University, Taipei 11031, Taiwan

**Keywords:** chemoresistance, small molecule, bioinformatics, cancer stem cells (CSCs), RV59, immune checkpoint

## Abstract

Tumor recurrence from cancer stem cells (CSCs) and metastasis often occur post-treatment in colorectal cancer (CRC), leading to chemoresistance and resistance to targeted therapy. *MYC* is a transcription factor in the nuclei that modulates cell growth and development, and regulates immune response in an antitumor direction by mediating programmed death ligand 1 (*PD-L1*) and promoting CRC tumor recurrence after adjuvant chemotherapy. However, the molecular mechanism through which *c-MYC* maintains stemness and confers treatment resistance still remains elusive in CRC. In addition, recent reports demonstrated that CRC solid colon tumors expresses C-X-C motif chemokine ligand 8 (*CXCL8*). Expression of *CXCL8* in CRC was reported to activate the expression of PD-L1 immune checkpoint through c-MYC, this ultimately induces chemoresistance in CRC. Accumulating studies have also demonstrated increased expression of *CXCL8*, matrix metalloproteinase 7 (*MMP7*), tissue inhibitor of metalloproteinase 1 (*TIMP1*), and epithelial-to-mesenchymal transition (*EMT*) components, in CRC tumors suggesting their potential collaboration to promote *EMT* and CSCs. *TIMP1* is *MMP*-independent and regulates cell development and apoptosis in various cancer cell types, including CRC. Recent studies showed that *TIMP1* cleaves *CXCL8* on its chemoattractant, thereby influencing its mechanistic response to therapy. This therefore suggests crosstalk among the *c-MYC*/*CXCL8*/*TIMP1* oncogenic signatures. In this study, we explored computer simulations through bioinformatics to identify and validate that the *MYC*/*CXCL8*/*TIMP1* oncogenic signatures are overexpressed in CRC, Moreover, our docking results exhibited putative binding affinities of the above-mentioned oncogenes, with our novel small molecule, RV59, Finally, we demonstrated the anticancer activities of RV59 against NCI human CRC cancer cell lines both as single-dose and dose-dependent treatments, and also demonstrated the *MYC*/*CXCL8*/*TIMP1* signaling pathway as a potential RV59 drug target.

## 1. Introduction

Colorectal cancer (CRC) is the 3rd most common cancer [1,2], with approximately 1.8 million cases diagnosed in both males and females in 2020 [3] and second in terms of death rates globally [4]. Approximately 25% of CRC patients exhibites metastatic disease at the diagnostic stage [5]. Current advanced treatment of CRC includes surgery and chemotherapy [6]. Despite these therapeutic efforts, huge challenges remain with regards to treatment responses [7], accumulation of tumor recurrence from cancer stem cells (CSCs), and metastasis often occurs post-treatment, which lead to chemoresistance and resistance to targeted therapy [8,9]. CSCs are able to self-renew and differentiation into heterogeneous cancer cells, subsequently promoting tumor initiation, proliferation, and recurrence [10,11]. Therefore, these limitations underscore the urgency of developing novel effective treatments. Identifying oncogenes responsible for stemness and chemoresistance in CRC is a significant route to new and effective therapeutic interventions [12]. *MYC* is a transcription factor in nuclei that modulates cell growth and development, and *MYC* overexpression promotes CRC development and progression [13]. *c-MYC* is a member of the *MYC* family, which is commonly expressed in CRC, and regulates cell proliferation, apoptosis, transformation, and therapeutic resistance. In CRC, *c-MYC* was demonstrated to regulate gene expression which plays an important role as regulator of epithelial stem cells in colon tissue [14,15]. In particular, the activation of *Wnt*/*β-catenin*, which is reported to play a major role as promoter of CSCs, serves as a transcriptional factor for *c-MYC* in the nucleus, which ultimately regulates major functions of CSCs. Therefore, *c-MYC* serves as one of the core stem cell markers [16,17].

Additionally, high expression levels of *c-MYC* were demonstrated to regulate antitumor immune responses, by mediating programmed death ligand 1 (*PD-L1*) and cluster of differentiation 47 (*CD47*) [18], as well as promoting CRC tumor recurrence after 5-fluorouracil (5-FU)-based adjuvant chemotherapy [19]. Several studies showed that when upregulated, *c-MYC* activates mitogen-activated protein kinase (*MAPK*) and rat sarcoma (*RAS*) signaling, which eventually alters the response to oxaliplatin or vorinostat treatment in breast cancer and CRC patients [20,21]. In addition, *c-MYC* was also reported to be one of the major stem cell markers in CRC. However, the molecular mechanism through which *c-MYC* maintains stemness and confers treatment resistance still remains elusive in CRC [19,22]. This therefore illustrates the need for further exploration of its regulatory mechanisms [23,24]. Recent reports demonstrated that solid colon tumors express different kinds of chemokines, including a member of the CXC chemokine family, C-X-C motif chemokine ligand 8 (*CXCL8*), also known as interleukin *(IL)-8* [7,25], In addition, *CXCL8* functions as an inflammatory cytokine secreted by immune cells, such as macrophages within the tumor microenvironment (TME) [26,27]. *CXCL8* expression in CRC was reported to activate the expression of the immune checkpoint *PD-L1* through *c-MYC* [28]. Moreover, *CXCL8* was revealed to play a significant role by promoting tumor formation, progression, and invasion via activating phosphoinositide 3-kinase (*PI3K*) signaling, which subsequently phosphorylates protein kinase B (*PKB*) also known as (*AKT*) and *MAPK* in breast cancer and CRC [29,30]. In addition, recent reports gradually showed that an increased level of *CXCL8* mediates tumor metastasis, stemness, and ultimately induces chemoresistance in CRC [31,32].

Furthermore, study evidence from a microarray analysis also outlined significant differences in several oncogenes between tumor tissues and normal tissues. Interestingly, *CXCL8*, *MMP7*, tissue inhibitor of metalloproteinase 1 (*TIMP1*), and components of the epithelial-to-mesenchymal transition (*EMT*) were found to be highly expressed in CRC tumors [33]. This suggests that *CXCL8*/*MMP7*/*TIMP7* oncogenes may collaborate and promote the EMT and colon stem cells [34,35]. *TIMP1* is matrix metalloproteinase (MMP)-independent and regulates cell development and apoptosis in various cancer cell types, including CRC [36]. Recent studies showed that *TIMP1* cleaves to *CXCL8* on its chemoattractant, thereby influencing its mechanisms and responses to therapy [37]. *TIMP1* was also reported to participate in various cell functions including proliferation and survival, leading to reduced sensitivity to chemotherapy in colon cancer [37,38,39]. Increased expression of *TIPM1* was shown to be associated with poor clinical outcomes in CRC patients compared to normal samples [40]. This therefore suggests crosstalk among *c-MYC*/*CXCL8*/*TIMP1* oncogenic signatures. In the current study, we used computation studies through a bioinformatics analysis to identify and validate expressions of the *c-MYC*/*CXCL8*/*TIMP1* signaling pathway in CRC, and used in silico molecular docking to evaluate potential interactions of the RV59 with *MYC*/*CXCL8*/*TIMP1* signaling. RV59 is a small molecule, a derivative of EGFR inhibitor osimertinib derivative [41,42], which was recently synthesized in our laboratory. The anticancer effects of the RV59 was evaluated using the US National Cancer Institute (NCI)-60 colon cancer cells to single-dose and dose-dependent treatments with RV59 [43].

## 2. Material and Methods

### 2.1. Identifying Molecular Targets and Therapeutic Classes of RV59

To predict potential drug targets of RV59, we used computer-based Prediction of Biological Activity Spectra (PASS) (http://way2drug.com/PassOnline/) web resources to predict the spectrum of interactions for known protein kinase inhibitors [44]. Additionally, we applied the Developmental Therapeutics Program (DTP)-COMPARE, a public data-mining website, to predict molecular targets and detect if RV59 retained activity similar to NCI synthetic compounds and standard agents. Herein, we used 50% growth inhibition (GI_50_), which is the IC_50_ as an endpoint, and National Safety Code (NSC) number (763967) as a delimiter [45].

### 2.2. Gene Expression Microarray Data Extraction

In total, three gene expression profiles (GSE41328, GSE44861, and GSE74602) were extracted from the Gene Expression Omnibus (GEO; https://www.ncbi.nlm.nih.gov/geo/). The GEO database, a free public genomic database, stores array data and sequence data. GSE74602 included 10 CRC cancer samples and 10 matched adjacent normal samples [46]; GSE44861 included 56 colon cancer tissues and 55 adjacent noncancerous tissues [47]; and GSE74602 contained 30 CRC carcinoma tissues and 30 adjacent normal cancer tissues [48].

### 2.3. Validation of MYC/CXCL8/TIMP1 Expression Levels in CRC

Expression levels of the *MYC*/*CXCL8*/*TIMP1* oncogenes were analyzed with UALCAN (http://ualcan.path.uab.edu/), an open-access public online tool for analysis of The Cancer Genome Atlas (TCGA) [49]. Expression levels of *MYC*/*CXCL8*/*TIMP1* in CRC samples (red) were compared to adjacent normal samples (blue), with *p <* 0.05 indicating statistical significance. Furthermore, we used GECO, a gene expression correlation analytical tool, which distinguishes two expression datasets into positive and negative correlations [50], with positive Pearson correlation coefficients and *p* < 0.05 as statistically significant.

### 2.4. Protein-Protein Interaction (PPI) Network, Gene Ontology (GO), and Kyoto Encyclopedia of Genes and Genomes (KEGG) Pathway Analyses

Protein interactions were analyzed using the STRING tool (https://string-db.org/) to construct a PPI clustering network [51]. A confidence score >0.7 was considered most significant. The GeneMenia tool (https://genemania.org/) and cytoscape software (version 3.8.2) were used to build gene interactions and PPI networks respectively, and from analysis of the results, the interactive networks were based on gene co-expression, co-localization, genetic interactions, and various pathways involved within the network. The database for annotation, visualization, and integrated discovery (DAVID), (https://david.ncifcrf.gov/.jsp), was used to analyze enriched GO including biological processes and molecular functions involved, with the criterion set to *p* < 0.05.

### 2.5. Interpretation of Gene Co-Expression in MYC/CXCL8/TIMP1 Genes Network

Interpretation of gene expression network was analyzed using the Network Analyst 3.0 tool (https://www.networkanalyst.ca/) [52], a comprehensive analytical visual platform which integrates PPI networks and gene co-occurrence networks and interprets gene expression networks. Herein, we used Enrichment Map, a sub-tool of network analyst, with a Bipartite view, to determine the enrichment of coexpressed genes in KEGG pathways [53].

### 2.6. Correlation Analysis of MYC/CXCL8/TIMP1 Expressions and Tumor Infiltration Levels

Correlations between *MYC*/*CXCL8*/*TIMP1* expressions and tumor infiltration levels were analyzed with the Tumor Immune Estimation Resource (TIMER) (https://cistrome.shinyapps.io/timer/), an online computational tool used to analyze the nature of tumor immune interactions across a variety of cancers [54]. Herein, we determined correlations of *MYC*, *CXCL8*, and *TIMP1* with a set of gene markers of immune infiltration cells including CD8+ T cells and macrophages (with *p* < 0.05). Furthermore, we applied the SCNA statistical module, a sub-tool of TIMER to identify the abundances of tumor infiltrates including CD8+ T cells, macrophages, and dendritic cells (DCs) in colon cancer. The infiltration level was compared to the normal level using a two-sided Wilcoxon rank-sum test.

### 2.7. Drug Sensitivity Analysis of MYC/CXCL8/TIMP1 Oncogenes

To determine the correlation between *MYC*/*CXCL8*/*TIMP1* oncogenes and drug sensitivity of the genomics of drug sensitivity in cancer (GDSC) top 30 drugs in pan-cancer, we used the Gene Set Cancer Analysis (GSCA), a web-based tool used to analyze differentially expressed genes (DEGs) and correlation to drug sensitivity [55]. All the drugs approved by the Food and Drug Administration (FDA) were displayed in this analysis.

### 2.8. Molecular Docking of Protein-Ligand Interactions

In silico molecular docking is a technique used to predict protein-ligand interactions, and this includes detection of the position and orientation of the ligand [56]. In order to evaluate the strength of interactions of RV59 with predicted and selected target genes from PASS and the DTP-compare algorithm, we performed molecular docking of RV59 with the *MYC*/*CXCL8*/*TIMP1* oncogenes. Crystal structures of *MYC* (PDB: 6G6K), *CXCL8* (PDB:5D14), and *TIMP1* (1UEA), were downloaded from the Protein Data Bank (PDB). The 3D structure of RV59 was constructed with the Avogadro molecular visualization tool [57], while the standard inhibitor 3D structures for vorinostat (CID:5311), with a molecular weight (MW: 264.32 g/mol) with molecular formula (C_14_H_2_ON_2_O_3_), repaxicin (CID:9838712), with a Mw of (283.39 g/mol and molecular formula (C_14_H_21_NO_3_) and ilomastat (CID: 132519), with a Mw of (388.5 g/mol and molecular formula (C_20_H_28_N_4_O_4_), were all retrieved from PubChem as SDF files. For further processing, we used PyMol software (https://pymol.org/2/) to visualize the ligands and convert them into PDB file format, and these files were subsequently converted into PDBQT format using the computer-based software, autodock (http://autodock.scripps.edu/resources/adt). For visualization and interpretation, docking results were analyzed using Discovery Studio [58].

### 2.9. Statistical Analysis

Pearson’s correlations were used to assess correlations of *MYC*/*CXCL8*/*TIMP1* expressions in CRC cancer types. The statistical significance of differentially expressed genes (DEGs) was evaluated using the Wilcoxon test. * *p* < 0.05 was accepted as being statistically significant.

## 3. Results

### 3.1. MYC/CXCL8/TIMP1 Oncogenes Are Potential Drug Targets for RV59

We explored computer-based PASS and DTP-COMPARE, a public data-mining website to predict molecular targets and detect if RV59 retained activities similar to NCI synthetic compounds and standard agents. Similarities between our test compound and correlated drugs are presented as ‘‘Pearson’s Correlation Coefficient” values, where a value of +1 indicates perfect positive correlation and a value of -1 indicates negative similar correlation. Herein, we used GI_50_ as an endpoint and the NSC number as a delimiter [45], and identified several RV59 druggable genes. Interestingly, *MYC*, *CXCL8*, and *TIMP1* were among the genes displayed by the prediction tools (Table 1). In addition to these finding, the activities of RV59 on the identified genes, shown by the PASS tool, were classified according to “probability to be active” (Pa) as opposed to “probability to be inactive” (Pi) values, based on the analyzed results (all Pa values were greater than Pi values), indicating the inhibitory activities of the compound on *MYC* and *CXCL8*, among other identified activities (Table 2).

### 3.2. RV59 Passed the Required Drug-Likeness Criteria

RV59 small molecule, is a derivative of EGFR inhibitor osimertinib (CID: 71496458), which was recently synthesized in our laboratory. Using swissADME, a free web tool to evaluate pharmacokinetics, drug-likeness, and medicinal chemical friendliness of small molecules [47], we assessed the absorption, distribution, metabolism, and excretion (ADME) of RV59. Bioavailability radar displaying all six physical properties of RV59 showed that the compound successfully met the minimum requirements of drug likeness (Figure 1). The criteria are based on the Mw of a compound (Mw recommended value ≤ 500 g/mol), flexibility (number of rotations: recommended value ≤ 10), solubility (log S (ESOL) recommended value 0–6), saturation (fraction Csp3 recommended value ≤ 1), polarity (TPSA recommended value ≤ 140 Å^2^), and lipophilicity (XLOGP3 recommended value −0.7–5). Moreover, pharmacokinetics (PKs), drug-likeness, and medicinal chemical properties of RV59 (Table 3) indicated that RV59 had good synthetic accessibility of 2.95; this was evaluated according to the range from 1 (easy to make) to 10 (difficult to make). The compound also passed the criteria for Ghose, Veber (GSK), Egan (Pharmacia), and muegge and the Lipinski (Pfizer) rule-of-five of drug likeness and drug discovery. The bioavailability of the compound based on gastrointestinal absorptivity (GIA) indicated a score of 0.55 (55%) which indicates acceptable PK properties.

### 3.3. Identification of Differentially Expressed Genes (DEGs) in CRC

DEGs from the microarray dataset were shown by identifying expressed genes between colon cancer samples and normal samples tallied from separate studies. Results showed that 536, 144, and 607 DEGs were respectively obtained from the GSE41328, GSE44861, and GSE74602 datasets. Eighty-five genes overlapped in these datasets as demonstrated in a Venn diagram (Figure 2A). Red and blue dots in the heatmap and volcano plots respectively represent upregulated and downregulated genes (Figure 2B–D). The expression density curve demonstrated that the expression scales of the three databases were between 0 and 16. Expression values of gene data in GSE41328 and GSE74602 were concentrated around 5 following standardization.

### 3.4. Validation of MYC/CXCL8/TIMP1 Expression Levels in CRC

Expression levels of the *MYC*/*CXCL8*/*TIMP1* oncogenes were analyzed using UALCAN (http://ualcan.path.uab.edu/), a comprehensive open-access public online tool for analyzing cancer data [49] Expression levels of the *MYC*/*CXCL8*/*TIMP1* oncogenes in CRC primary tumor samples (red) were compared to adjacent normal samples (blue), with *p <* 0.05 considered statistically significant (Figure 3A–C). Interestingly, the analytical results showed high expressions of the *MYC*/*CXCL8*/*TIMP1* oncogenes in tumor samples compare d to normal samples. Furthermore, we used GECO, a gene expression correlation analytical tool, which displayed positive correlations of *MYC* with *CXCL8*, *MYC* with *TIMP1*, and *TIMP1* with *CXCL8*. Positive Pearson correlation coefficients and *p* < 0.05 indicated statistical significance (Figure 3D–F).

### 3.5. PPI Network and GO and KEGG Pathway Analysis

A functional interaction analysis was performed with STRING (https://string-db.org/) to construct a PPI clustering network [51]. A confidence score of >0.7 was considered most significant. The GeneMenia tool (https://genemania.org/) and cytoscape software were used to build the PPI network. From analysis of the results, interactive networks were based on gene co-expression, co-localization, genetic interactions, and various pathways involved within the network. In short, we identified interactions of *CXCL8* with *MYC*, *TIMP1*, *MMP9* and *ACKR*; of *MYC* with *CXCL8*, *TIMP1*, *MMP9*, *CDK4*, *CCND1*, *RIOX2*, *BCAT* and *DDX18*; and of *TIMP1* with *MYC*, *CXCL8*, *MMP9*, *CCNA2*, *JUND*, *and STAT3* (Figure 4A,B). The DAVID database (https://david.ncifcrf.gov/.jsp) was used to analyze the enriched GO including involved biological processes and biological pathways, with the criterion set to *p* < 0.05 (Figure 4C,D).

### 3.6. MYC/CXCL8/TIMP1 Gene Co-Expression and Functional Enrichment Analysis

Gene co-expressions displayed the most factors which contributed to functional interactions. Herein, we determined the signaling network and KEGG pathway functional enrichment analysis, and found that the *MYC*, *IL10*, *TP53*, *CXCL8*, *TIMP1*, and *CDK2* genes were co-expressed and were the most enriched (Figure 5A). Moreover, the KEGG pathway enrichment analysis showed that co-expressed genes exhibited enrichment in the thyroid hormone signaling pathway, bladder cancer, cellular senescence, cell cycle, and HTLC-1 infections (Figure 5B).

### 3.7. MYC/CXCL8/TIMP1 Expressions Were Correlated with Immune Cell Infiltration in Both Cancer and Normal Tissues

To identify associations of *MYC*/*CXCL8*/*TIMP1* expressions with selected immune cells, we applied a correlation analysis between the above-mentioned oncogenes and immune infiltration cells (CD8+ and macrophages), where markers were adjusted by purity. As expected, results showed correlations of immune cell markers in colorectal adenocarcinoma (COAD), specifically CD8+ T cells and M2 macrophages (Figure 6A–C), with *p* < 0.05 considered significant. Expressions of MYC/*CXCL8*/*TIMP1* were also found to be correlated with infiltrating levels of CD8+ T cells, macrophages, and DCs, and red represents the most significant positive correlations with high amplification, while blue represents negative correlations (Figure 6E–G). The infiltration level was compared to the normal level using a two-sided Wilcoxon rank-sum test.

### 3.8. Drug Sensitivity Analysis of MYC/CXCL8/TIMP1 Oncogenes

To determine the drug sensitivity of *MYC*, *CXCL8* and *TIMP1*, we used the GSCA tool to analyze the drug response (Figure 7). The correlation coefficients analysis, shows that upregulated gene expression is associated with drug resistance. From our analysis of results, we identified increased mRNA expression levels of *MYC*, *CXCL8* and *TIMP1* (indicated in orange bubbles), to be less sensitive to the drugs. Interestingly, high expression levels of *MYC*, *TIMP1* and *CXCL8* gene signatures, were shown to be resistance to Bx-912 (*PDK-1* inhibitor) [59], navitoclax (Bcl-2 inhibitor) [60], vorinostat (*HDAC* and MYC inhibitor) [61,62] and tubastatin A (*HDAC* inhibitor) [63] among other FDA approved drugs. 

### 3.9. Docking Results Displayed Strong Binding Energies between RV59 and the MYC/CXCL8/TIMP1 Oncogenes

Results from the in silico molecular docking analysis revealed unique binding affinities of the RV59 compound with selected target genes obtained from the PASS online prediction tool and DPT−COMPARE algorithm. Gibbs free binding energy results were obtained as follows: −7.6 kcal/mol for *MYC*, −7.7 kcal/mol for *CXCL8*, and −6.9 kcal/mol for *TIMP1*. In addition, a visualization analysis showed that docking of the small molecule (ligand) displayed shorter binding distances with the proteins (receptors) of 2.49 Å for *MYC*, of 2.03 Å for *CXCL8*, and of 2.51, 3.14, 3.4, and 3.29 Å for *TIMP1* (Figure 8A,B). Moreover, we compared results obtained from RV59 docking with standard inhibitors of *MYC* (vorinostat), *CXCL8* (reparixin), and *TIMP1* (ilomastat). Interestingly, the standard inhibitors exhibited lower binding energies for *MYC* (−6.3 kcal/mol) and *CXCL8* (−6.3 kcal/mol), with the exception of *TIM1* (7.4 kcal/mol), which showed a slightly higher binding energy compared to the RV59 compound (Figure 9A–C). In addition, several interactions were identified for the ligand and protein complex, and these interactions included amino residues, a high number of conventional hydrogen bonds, van der Waals forces, carbon hydrogen bonds, Pi anions, Pi-sigma, Pi-Pi stacked, and amide Pi-stacked as shown in Table 4.

### 3.10. Expressions of MYC/CXCL8/TIMP1 Oncogenes across Colon Cancer Cell Lines

To identify expression levels of the *MYC*/*CXCL8*/*TIMP1* oncogenes in different CRC cell lines, we explored the expression database web tool, https://www.ebi.ac.uk/gxa/home [64]. We used the RNA-sequence data of 675 commonly used human cancer cell lines for each gene. Results showed increased expression levels of the *MYC*/*CXCL8*/*TIMP1* oncogenes across colon cancer cell lines (Figure 10). 

### 3.11. RV59 Displayed Anti-Proliferative and Cytotoxic Effects in NCI60 Human Colon Cancer Cell Lines

RV59 showed anticancer activities against NCI human colon cancer cell lines (Figure 10). A single-dose treatment was administered at an initial dose of 10 μM, growth percentages revealed RV59 cytotoxic effects on the COLO 205, HCC-2998, and HCT-15 cell lines, and anti-proliferative activities on the HCT-116, HT29, KM12, and SW-620 cell lines (Figure 11A). Since the compound showed potential anticancer effects at an initial dose of 10 μM, further dose-dependent investigations of the compound were performed to measure GI_50_, tumor growth inhibition (TGI), and 50% lethal concentration (LC_50_) values. Results showed potential anti-proliferative effects in a dose-dependent manner (Figure 11B,C). Moreover, the in vitro IC_50_ results ranged 0.18–1.85 μM in the colon cancer cell lines, with HCT-15 being the more responsive at 0.18 μM, followed by HCT116 (at 0.28 μM, SW-620 at 0.29 μM, HT-29 at 0.47 μM, KW12 at 0.84 μM, and COLO 205 (at 1.15 μM, with HCC-2998 at 1.85 μM showing the least responsiveness compared to the aforementioned cell lines (Figure 11D).

## 4. Discussion

Advanced chemotherapy and targeted therapies still offer limited prolonged overall survival in CRC patients. One of the main causes of poor prognoses in patients is resistance to these therapeutic interventions [65]. Moreover, the molecular mechanisms through which cancer escapes chemotherapy and targeted therapy still remain elusive, mainly due to colon cancer’s heterogenic properties. CRC is often diagnosed at a later, advanced stage, with distant metastasis present in most cases [66]. Therefore, there is a need for novel and effective targeted therapies, to improve patient’s clinical outcomes and resistance in CRC patients. Protein kinases dysregulations have been reported to drive cancer, due to its association with genetic alterations, such as overexpression and mutations. Therefore, they have become pharmaceutical targets over the years [67]. RV59 is a novel small molecule derived in our lab from *EGFR* inhibitor osimertinib [68]. Our previous studies evaluated the anticancer activities of 20 nitrogen-substituted anthra[1,2-c][1,2,5]thiadiazole-6,11-dione derivatives, on cytoplasmic nuclear location sequence (NLS)-mutated *Nrf2*-transfecte, which promotes CRC tumor invasion and resistance to 5-flurouracil (5-FU) chemotherapy. Among those 20 compounds, RV59 was more effective, overcame *cNrf2-*mediated chemoresistance, and suppressed tumor growth in colon cancer cells [69]. However, PK and toxicity analyses were not performed in that study. Further, in another study from our lab, we showed that RV59, exhibited a broad-spectrum of cytotoxicity against various cancer cells, and interestingly, the compound displayed less cytotoxic effects as compared to chemotherapeutic agent, doxorubicin in normal tissues [70].

Herein, we used a bioinformatics simulation study to further expand on that previous study; however, the present study focused more on identifying and validating oncogenes associated with chemoresistance and alteration of oxaliplatin or doxorubicin treatment responses in advanced CRC. Studies showed that accumulation of tumor recurrence from cancer stem cells (CSCs), and metastasis often occur post-treatment in CRC, which leads to therapeutic resistance [65,71]. Using computer-based PASS and DTP-COMPARE drug target predictive tools, we identified *MYC*/*CXCL8*/*TIMP1* as target genes for RV59; in addition, an analysis of the results also predicted that all probability active (Pa) values were greater that all predicted probability inactive (Pi) values, indicating the inhibitory or antagonistic activities of RV59 on *MYC* and *CXCL8*, among other identified activities. In the early stages of drug discovery and development, the disposition of a compound is assessed in terms of its absorption, distribution, metabolism, and excretion (ADME), with the final goal of identifying potential effectiveness of a medicine for patients [72]. After assessing the ADME of RV59, bioavailability radar displaying all six physical properties of the compound, showed that the compound successfully met the minimum requirements of drug-likeness, with an Mw of 352.41 g/mol, flexibility (rotations = 4), solubility (log S (ESOL) = −3.74), saturation (fraction Csp3 = 0.22), polarity (TPSA = 103.43 Å^2^), and lipophilicity (XLOGP3 = 2.76), and all the values were within recommended values as described in Figure 1. The compound also passed the criteria for Ghose, Veber (GSK), Egan (Pharmacia), and muegge and the Lipinski (Pfizer) rule-of-five of drug likeness and drug discovery. The bioavailability of the compound based on GIA indicated a score of 0.55 (55%), which indicates acceptable PK properties. Moreover, we explored the UALCAN bioinformatics tool, and validated increased expressions of *MYC*/*CXCL8*/*TIMP1* oncogenic signatures in CRC primary tumors and compared them to adjacent normal samples. The genes also exhibited positive correlations among each other with *p* < 0.05 indicating statistical significance.

In a further analysis, we determined interactions of the *MYC*/*CXCL8*/*TIMP1* oncogenes at the protein level. Interestingly, the PPI network analysis from two independent databases showed interactions of all of the proteins with each other, as well as enrichment in GO terms, including biological processes and biological pathways involved in CRC, with *p* < 0.05 indicating statistical significance (Figure 4). *MYC* plays significant roles in tumorigenesis and therapeutic resistance [15]. In 2019, Han et al. demonstrated that *MYC* induces immunogenic cell death of tumor cells, which leads to increased *T cell* infiltration and upregulation of the *PD-L1* immune checkpoint protein in the TME [73].

Additional studies also showed that inhibition of programmed cell death protein 1 (*PD1*) by pembrolizumab enhanced regulation of cytotoxic *T-cell* tumoricidal activities, which ultimately leads to increased expression levels of tumor-enhanced *CXCL8*, which subsequently induces infiltration of tumor-associated macrophages (M2) in the immune system [74,75]. Others showed that the *TIMP1* gene derived from tumor cells creates a metastatic niche, to which circulating tumor cells cling and promote CRC metastasis [76]. Hence, these finding validated the potential contribution of the *MYC*/*CXCL8*/*TIMP1* oncogenic signatures to chemoresistance and resistance to targeted treatment. To identify associations of *MYC*/*CXCL8*/*TIMP1* gene expressions with selected immune cells in CRC, we applied a correlation analysis between the above-mentioned oncogenes with immune infiltration cells (*CD8+* and macrophages), where markers were adjusted by purity. As expected, *MYC*/*CXCL8*/*TIMP1* gene expression levels displayed associations with tumor purity and were positively correlated with infiltrating levels of CD8+ T cells and macrophages (*p* < 0.05), and were correlated with the abundances of tumor infiltrates including CD8+ T cells, macrophages and DCs in CRC (Figure 6).

We further predicted protein-ligand interactions using molecular docking between RV59 and the *MYC*/*CXCL8*/*TIMP1* gene complex, and discovered that RV59 displayed strong binding energies to *MYC* (−7.6 kcal/mol), *CXCL8* (−7.7 kcal/mol), and *TIMP1* (−6.9 kcal/mol), and short binding distances with these protein (receptors): *MYC* (2.49 Å), *CXCL8* (2.03 Å), and *TIMP1* (2.51, 3.14, 3.4, and 3.29 Å). These results were compared to standard inhibitors of *MYC* (vorinostat), *CXCL8* (reparixin), and *TIMP1* (ilomastat). Interestingly, the inhibitors exhibited lower binding energies for *MYC* (−6.3 kcal/mol) and *CXCL8* (−6.3 kcal/mol), with the exception of *TIMP1* (7.4 kcal/mol) compared to RV59. Moreover, RV59 showed anticancer activities against NCI human colon cancer cell lines with single-dose treatment of 10 μM, toward the COLO 205, HCC-2998, and HCT-15 cell lines, and anti-proliferative activities toward the HCT-116, HT29, KM12, and SW-620 cell lines, in dose-dependent manners. measured Results of measuring the GI_50_, TGI, and LC_50_ levels showed potential anti-proliferative effects in dose-dependent manners (Figure 11B,C). Moreover, the in vitro IC_50_ results ranged 0.18–1.85 μM on colon cancer cell lines, with HCT-15 cells being the most responsive at 0.18 μM, followed by HCT116 at 0.28 μM, SW-620 at 0.29 μM, HT-29 at 0.47 μM, KW12 at 0.84 μM, and COLO 205 at 1.15 μM, with HCC-2998 cells at 1.85 μM showing the least responsiveness compared to the aforementioned cell lines. This suggests that RV59 exhibits drug-like characteristics, and is a potential oral drug candidate. In summary, the findings from this study revealed the anticancer activities of RV59 in CRC, and highlight additional avenues for RV59 as a potential treatment for CRC chemotherapy and resistance to targeted therapy, particularly by targeting the *MYC*/*CXCL8*/*TIMP1* signaling pathway. The compound is currently being assessed for its therapeutic potential effectiveness in CRC and breast cancer both in vitro and in vivo in our laboratory.

## 5. Conclusions

In conclusion, we revealed the anticancer activities of RV59 against NCI human colon cancer cell lines both as a single dose and dose-dependent treatment, and also demonstrated the *MYC*/*CXCL8*/*TIMP1* signaling pathway, which is responsible for resistance to both chemotherapies and targeted therapies, to be a potential RV59 drug target. Moreover, the *in silico* molecular docking study exhibited putative binding affinities of RV59 with the above-mentioned oncogenes, which were even higher than the standard inhibitors of these genes. Currently in our lab, there are ongoing cell and animal experiments to evaluate the therapeutic effectiveness of RV59 in CRC and breast cancer.

## Figures and Tables

**Figure 1 cells-10-01970-f001:**
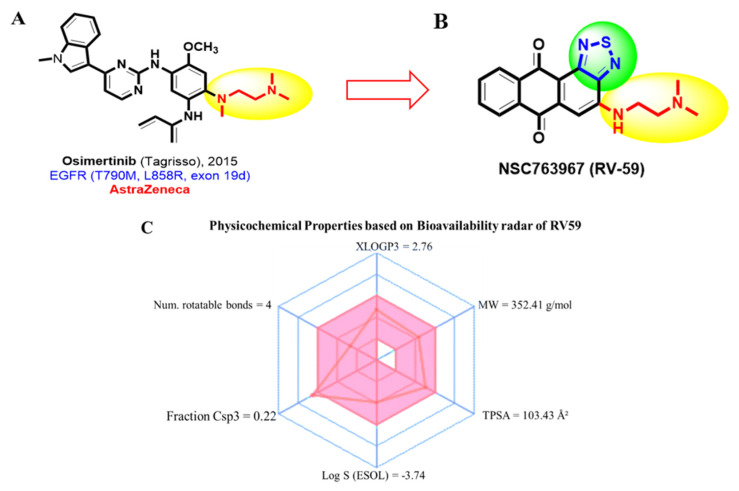
RV59 met the drug-likeness requirements. (**A**) RV59 small molecule, derivative of *EGFR* inhibitor (**B**) osimertinib (CID: 71496458). (**C**) Bioavailability radar displaying all six drug likeness physical properties of RV59; Mw of 352.41 g/mol, flexibility (rotations = 4), solubility (log S (ESOL) = −3.74), saturation (frac Table 3 = 0.22), polarity (TPSA = 103.43 Å^2^), and lipophilicity (XLOGP3 = 2.76).

**Figure 2 cells-10-01970-f002:**
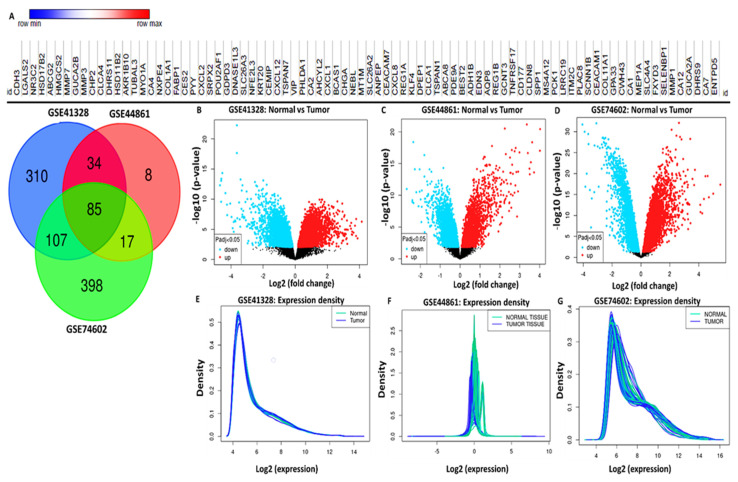
Identification of differentially expressed genes (DEGs) from the GSE41328, GSE44861, and GSE74602 microarray datasets. (**A**) Heat map and Venn diagram of 85 selected overlapping DEGs. (**B**–**D**) Volcano plots of DEGs from the three datasets with red and blue dots respectively representing upregulated and downregulated genes, with *p* < 0.05 and |logFC| ≥ 1 statistically significant between normal colon tissues and tumor tissues. (**E**–**G**) Expression density curve showing that expressions of the three databases ranged 0–16. Expression values of gene data in GSE41328 and GSE74602 were concentrated around 5 following standardization.

**Figure 3 cells-10-01970-f003:**
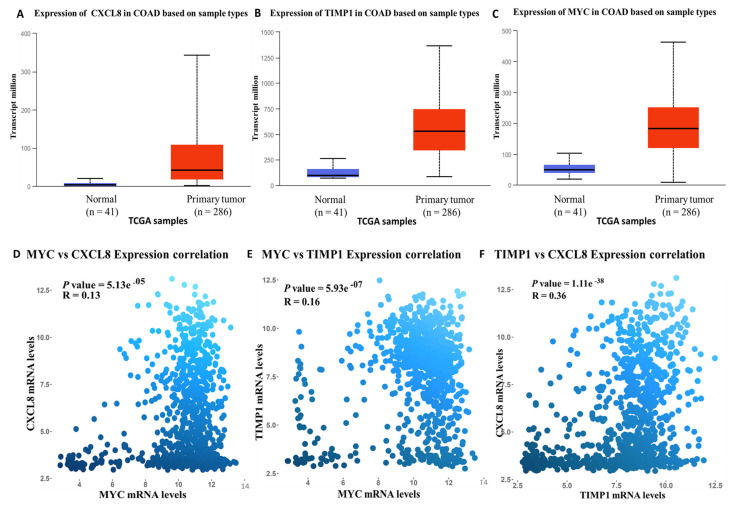
Increased expression levels of *MYC*/*CXCL8*/*TIMP1* signatures were associated with poor clinical outcomes in colon cancer. (**A**) *MYC*, (**B**) *TIMP1*, and (**C**) *MYC* expression levels in primary tumors compared to normal tissues, with *p* < 0.05 considered statistically significant. Correlation analysis showing positive correlations of (**D**) *MYC* with *CXCL8*, (**E**) *MYC* with *TIMP1*, and (**F**) *TIMP1* with *CXCL8*. Positive Pearson correlation coefficients and *p* < 0.05 indicated statistical significance.

**Figure 4 cells-10-01970-f004:**
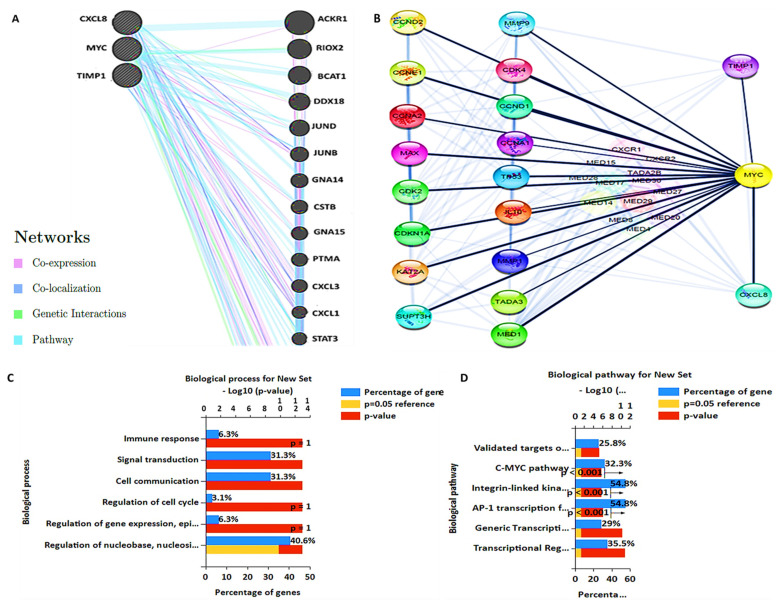
Protein−protein interactions displayed co−expression in the MYC/*CXCL8*/*TIMP1* network. (**A**,**B**) Interactive networks based on gene co-expression, co-localization, genetic interactions, and various pathways involved within the network. In short, we identified interactions of *CXCL8* with *MYC*, *TIMP1*, *MMP9*, *and ACKR*; of *MYC* with *CXCL8*, *TIMP1*, *MMP9*, *CDK4*, *CCND1*, *RIOX2*, *BCAT*, and *DDX18*; and of *TIMP1* with *MYC*, *CXCL8*, *MMP9*, *CCNA2*, *JUND*, and *STAT3*. (**C**,**D**) Enrichment of gene ontology (GO) including biological processes and biological pathways involved in colorectal cancer, with the criterion set to *p* < 0.05 as statistically significant.

**Figure 5 cells-10-01970-f005:**
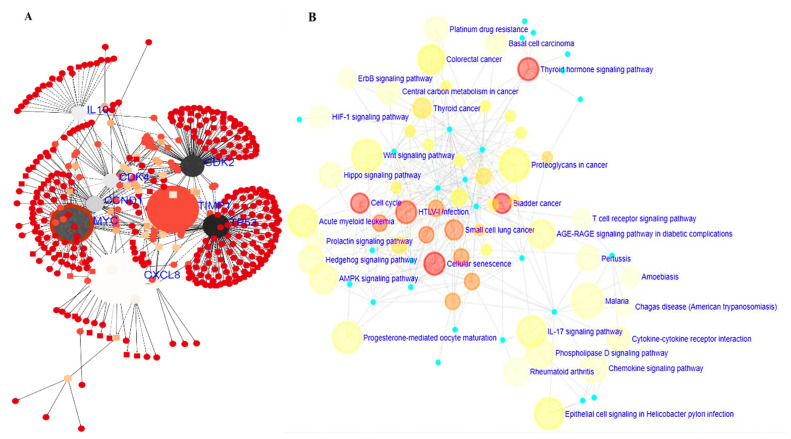
*MYC*/*CXCL8*/*TIMP1* oncogenes co-occurred and contributed to functional relationships. (**A**) Signaling network analysis, showing that co-expressions of the *MYC*, *IL10*, *TP53*, *CXCL8*, *TIMP1*, and *CDK2* genes were more enriched. (**B**) KEGG pathway enrichment analysis showed that co-expressed genes exhibited enrichment in the thyroid hormone signaling pathway, bladder cancer, cellular senescence, cell cycle, and HTLC-1 infections.

**Figure 6 cells-10-01970-f006:**
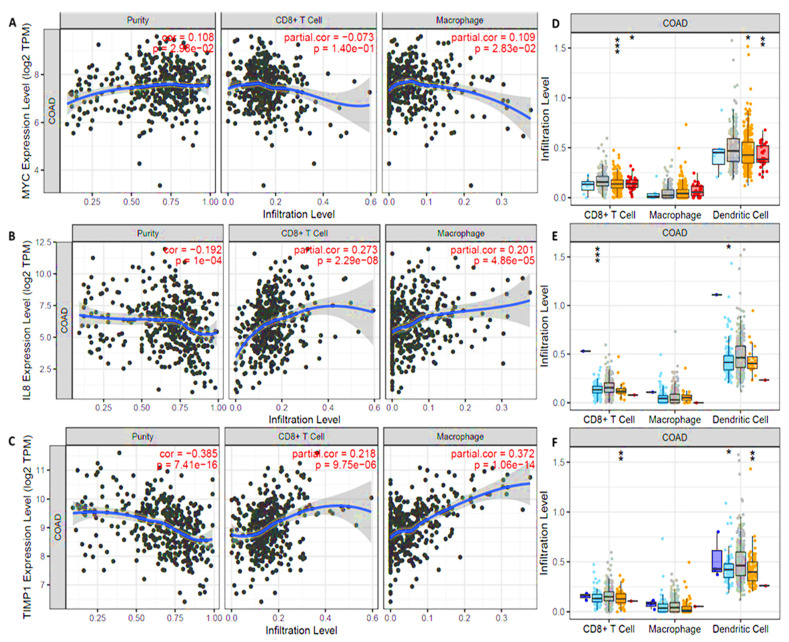
*MYC*/*CXCL8*/*TIMP1* expressions correlated with immune infiltrating cells in (COAD). (**A**) *MYC*, (**B**) *CXLC8*, and (**C**) *TIMP1* expression levels displayed associations with tumor purity and were positively correlated with infiltrating levels of CD8+ T cells and macrophages. *p* < 0.05 was considered statistically significant. (**D**) *MYC*, (**E**) *CXLC8*, and (**F**) *TIMP1* expressions were correlated with abundances of tumor infiltrates including CD8+ T cells, macrophages, and dendritic cells in colon cancer. Red represents significant positive correlations with high amplification, while blue represents negative correlations. The infiltration level was compared to the normal level using a two-sided Wilcoxon rank-sum test. *p*-value Significant Codes: 0 ≤ *** < 0.001 ≤ ** < 0.01 ≤ * < 0.05 ≤ . < 0.1.

**Figure 7 cells-10-01970-f007:**
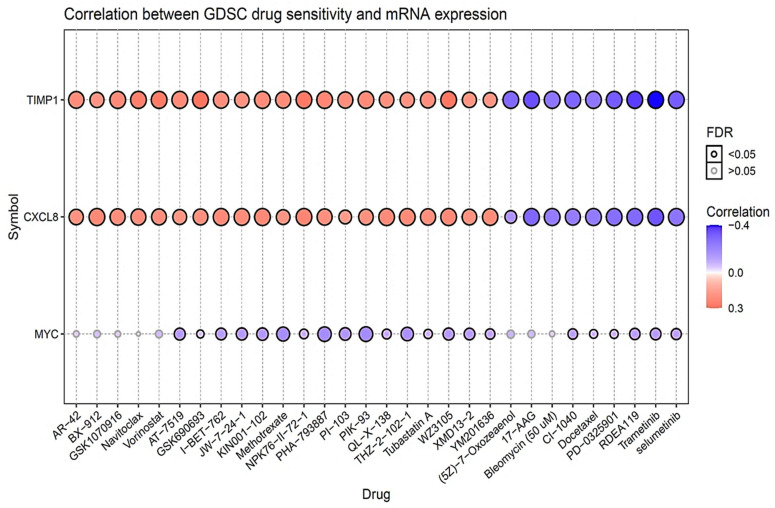
Drug sensitivity of *MYC*/*CXCL8*/*TIMP1 oncogenes* from GSCA. The figures show the correlation between genomics of drug sensitivity in cancer (GDSC) from FDA approved drugs. The positive Spearman correlation coefficient (orange bubbles), indicates that increased gene expression level is resistant to the drug, as compared to negative correlation shown in (blue), which indicates the sensitivity of the drug.

**Figure 8 cells-10-01970-f008:**
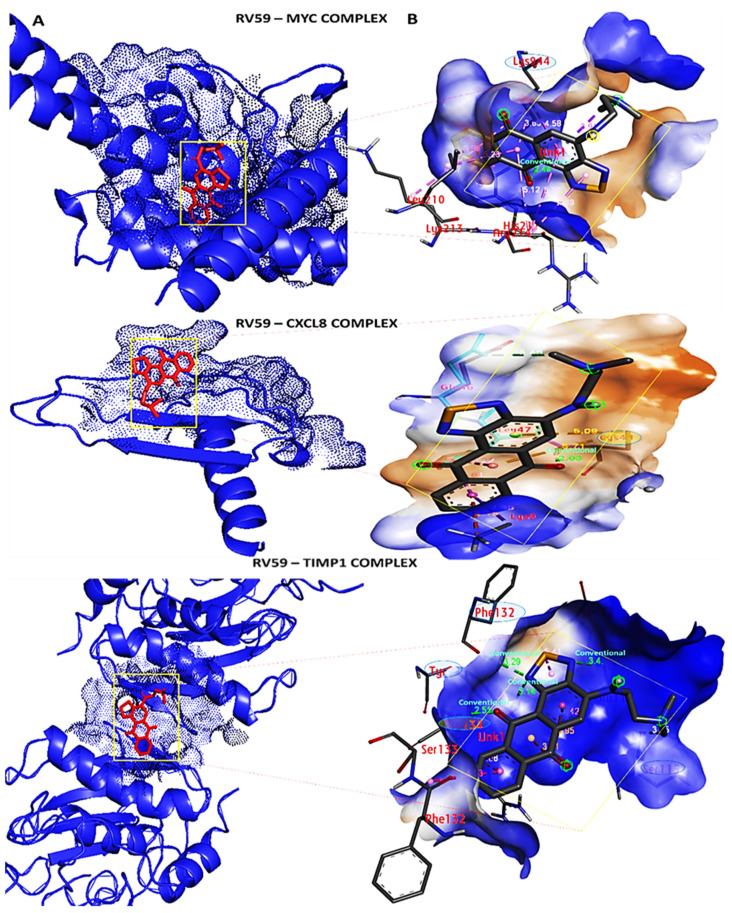
Molecular docking profiles of RV59 with the *MYC*/*CXCL8*/*TIMP1* oncogenic signatures. (**A**) 3D representation showing strong binding affinities of RV59 with *MYC* at −7.6 kcal/mol, *CXCL8* at −7.7 kcal/mol, and *TIMP1* at −6.9 kcal/mol. (**B**) 2D visualization of docking analysis of RV59 (ligand) displaying interactions between active side residues through conventional hydrogen bonding (green) with *LYS944* for *MYC*; *LYS48* for *CXCL8;* and *PHE132*, *TYR99*, *GLU118*, and *ARG134* for *TIMP1.* Short binding distances were also shown with the protein (receptors), *MYC* (2.49 Å), *CXCL8* (2.03 Å), and *TIMP1* (2.51, 3.14, 3.4, and 3.29 Å). Results were further stabilized by other interactions, including high amino acid residues, van der Waals forces, carbon-hydrogen bonds, Pi-anions, Pi-sigma, Pi-Pi stacked, and amide Pi-stacked as shown in the accompanying Table 4.

**Figure 9 cells-10-01970-f009:**
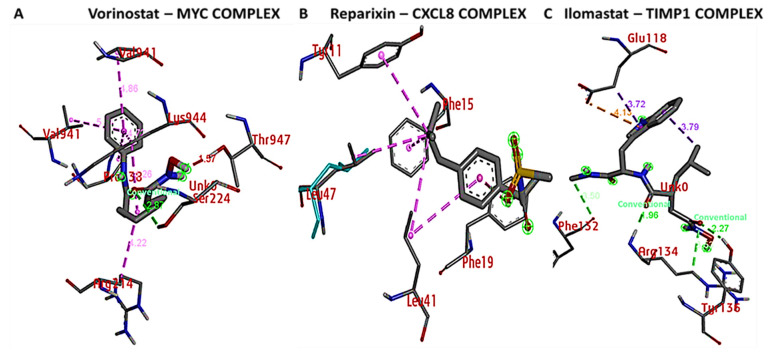
Docking profiles of the *MYC*/*CXCL8*/*TIMP1* oncogenes in complex with their respective inhibitors of vorinostat, reparixin, and ilomastat. (**A**) *MYC-*vorinostat complex displaying conventional hydrogen bonding with Gibbs free energy of 2.87 kcal/mol. (**B**) *CXCL8-*reparixin complex showing no conventional hydrogen bonding (green) and no putative binding energies. (**C**) *TIMP1*-ilomastat complex showing conventional hydrogen bonding with binding energies of 1.96 and 2.27 kcal/mol. These results suggest that RV59 displayed a more-interactive property with *MYC*/*CXCL8*/*TIMP1* oncogenic signatures compared to their standard inhibitors. Results were further stabilized by other interactions, including high amino acid residues, van der Waals forces, carbon hydrogen bonds, Pi-anions, Pi-sigma, Pi-Pi stacked, and amide Pi-stacked as shown in the accompanying Table 4.

**Figure 10 cells-10-01970-f010:**
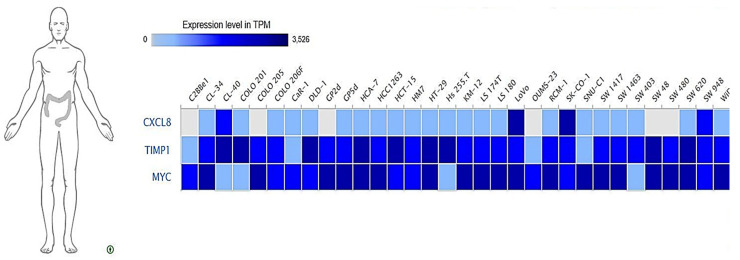
Shows increased expressions of *MYC/CXCL8/TIMP1* oncogenic signatures across a panel of colon cancer cell lines.

**Figure 11 cells-10-01970-f011:**
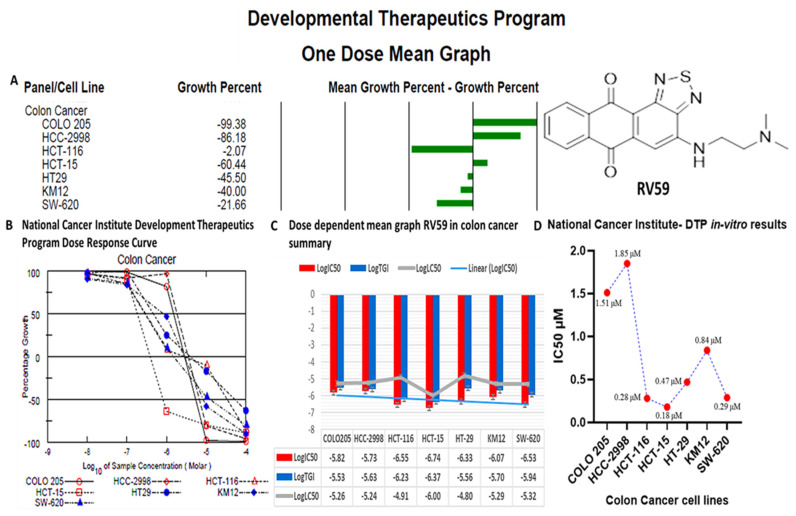
RV59 displayed anti-proliferative and cytotoxic effects in NC-I60 human colon cancer cell lines. (**A**) A single-dose treatment administered at 10 μM, revealed growth percentage cytotoxic effects of RV59 on the COLO 205, HCC-2998, and HCT-15 cell lines, and anti-proliferative activities on the HCT-116, HT29, KM12, and SW-620 cell lines. (**B**,**C**) Dose-dependent responses of colon cancer cell lines evaluated using the 50% growth inhibition (GI_50_), tumor growth inhibition (TGI), and 50% lethal concentration (LC_50_). (**D**) In vitro IC_50_ results ranged 0.18–1.85 μM on colon cancer cell lines, with HCT-15 being the most responsive at 0.18 μM, followed by HCT116 at 0.28 μM, SW-620 at 0.29 μM, HT-29 at 0.47 μM, KW12 (0.84 μM), and COLO 205 at 1.15 μM, with HCC-2998 showing the least responsiveness compared to the aforementioned cell lines at 1.85 μM.

**Table 1 cells-10-01970-t001:** Correlations of RV59 with NCI synthetic compounds and standard anticancer agents that share similar anticancer fingerprints and mechanisms.

		NCI-Synthetic Compounds			NCI-Standard Agents			ArrayCGH-Gray	
Rank	*r*	CCLC	Target Descriptor	*r*	CCLC	Target Descriptor	*r*	CCLC	Target Descriptor
1	0.6	50	3-Nitro-5-Formylisoxazole	0.51	58	Trimetrexate	0.27	57	MET
2	0.58	58	CI.49700	0.49	59	Methotrexate	0.27	57	CDK4
3	0.56	57	Tryptanthrin	0.46	58	Dichloroali lawsone	0.2	55	YES1
4	0.55	58	Tolonium Chloride	0.46	58	5HP	0.18	53	DRD3
5	0.54	57	Metoprine (USAN)	0.46	59	Cyclocytidine	0.13	55	TP53
6	0.54	45	Bafilomycin Deriv	0.42	59	Thioguanine	0.13	55	WNT1
7	0.53	53	Brilliant cresyl blue	0.41	58	Hycanthone	0.11	55	MYC
8	0.52	46	Bafilomycin Antibiotic	0.4	45	Tetraplatin	0.11	57	AKT1
9	0.52	41	Lapachol	0.4	58	Cytosine arabinoside	0.1	56	ABR
10	0.51	58	Piroctone olamine	0.38	58	Pyrazofurin	0.1	56	CCND1

*r*, Pearson’s correlation coefficient values range from −1 to +1 (the higher the positive number, the more positive correlation); CCLC, common cell lines count.

**Table 2 cells-10-01970-t002:** Prediction of Biological Activity Spectra (PASS) of the RV59 compound.

Pa	Pi	Activity
0.477	0.029	MYC inhibitor
0.375	0.166	Catenin beta inhibitor
0.302	0.139	MAP kinase stimulant
0.238	0.086	Antineoplastic (colon cancer)
0.113	0.058	Protein kinase (CK1) inhibitor
0.268	0.227	MAP kinase 8 inhibitor
0.210	0.174	Transcription factor NF kappa A inhibitor
0.143	0.110	Protein kinase (CK1) delta inhibitor
0.136	0.109	Acetylcholine M2 receptor antagonist
0.146	0.134	Chemokine (C-X-C motif) ligand 8 (CXCL8) antagonist

Pa > Pi, Pa, probability to be active; Pi, probability to be inactive.

**Table 3 cells-10-01970-t003:** Pharmacokinetics, drug-likeness, and medicinal chemistry of RV59.

Pharmacokinetics	GI Absorption (High)	BBB (Low)
Drug-likeness (Yes to all)	Lipinski, Ghose, Veber, Egan, Muegge	
Bioavailability score	55%	
Medical Chemistry	Synthetic accessibility:	2.95

Pharmacokinetics displayed high gastrointestinal absorptivity (GIA) and low blood-brain barrier (BBB) permeability; drug-likeness, the compound passed all the criteria for Ghose, Veber (GSK), Egan (Pharmacia), and muegge, and the Lipinski (Pfizer) rule-of-five of drug likeness and drug discovery. The bioavailability of the compound based on GIA indicated a score of 0.55 (55%) and synthetic accessibility of 2.95 evaluated according to the range from 1 (easy to make) to 10 (difficult to make).

**Table 4 cells-10-01970-t004:** Analytical summary table showing interactions of RV59 with the *MYC*/*CXCL8*/*TIMP1* oncogenic signatures compared the standard inhibitors of these genes.

RV59		Standard Inhibitotrs	
**RV59-MYC Complex** **(ΔG = −7.6 Kcal/mol)**		**Vorinostat-MYC Complex** **(ΔG = −6.3 Kcal/mol)**	
Type of interactions and number of bonds	distance of interacting Amino acids	Type of interactions and number of bonds	distance of interacting Amino acid
Conventional Hydrogen bond (1)	LYS944 (2.49 Å)	Conventional Hydrogen bond (1)	SER (2.87Å)
Pi sigma	LEU217	Van der Waals forces	LEU94, SER221, LUE225, LYS213, ASP220, VAL940
Pi-pi stacked	HIS217	Alkyl	ARG214, LYS944
Pi-Alkyl	ARG214,LYS213		
**RV59-CXCL8 Complex** **(** **ΔG = −7.7 Kcal/mol)**		**Reparixinx-CXCL8 Complex** **(** **ΔG = −6.3 Kcal/mol)**	
Type of interactions and number of bonds	distance of interacting Amino acids	Type of interactions and number of bonds	distance of interacting Amino acids
Conventional Hydrogen bond (1)	CYS47 (2.03 Å)	Van der Waals forces	ASP43, ARG45
Carbon hydrogen bond	GLU46	Pi-pi stacked	PHE19
Pi-cation	LYS9	Alkyl	LEU41, TYR11
Pi-Alkyl	LEU47	Pi-Alkyl	LEU47, PHE15
**RV59-TIMP1 Complex** **(ΔG = −6.9 Kcal/mol)**		**Ilomastat-TIMP1 Complex** **(** **ΔG = −7.4 Kcal/mol)**	
Type of interactions and number of bonds	distance of interacting Amino acids	Type of interactions and number of bonds	distance of interacting Amino acids
Conventional Hydrogen bond (4)	PHE132 (3.29 Å), TRY99 (3.41 Å), GLU118 (3.4 Å), ARG134 (2.51 Å)	Conventional Hydrogen bond (2)	ARG134, TYR136
Carbon hydrogen bond	SER133	Van der Waals	THR131, TYR99, GLU118, ASP114, GLU125, LYS122
Pi-cation	SER115	Carbon hydrogen bond	PHE132
Pi-Alkyl	PHE132	Pi-Anion	GLU118

## Data Availability

The datasets generated and/or analyzed in this study are available on reasonable request.

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
