# Peer review of "Comprehensive Omics Analysis of a Novel Small-Molecule Inhibitor of Chemoresistant Oncogenic Signatures in Colorectal Cancer Cell with Antitumor Effects"

_cells, 2021, doi:10.3390/cells10081970_

Round 1
Reviewer 1 Report
The present study investigates MYC/CXCL8/TIMP1 axis in colorectal cancer and potentially targeting that pathway by a novel small molecule inhibitor RV59. The authors have done a good job in explaining their methodology and results. Here are few comments that need to be addressed: 1) The title has a grammatical error: It should rather be "Comprehensive Omics Analysis OF a Novel Small-Molecule Inhibitor of Chemoresistant Oncogenic Signatures in Colorectal Cancer Cell with Antitumor Effects. 2) Authors should modify their introduction to provide latest stats for CRC also provide references for information on c-myc, its role in cancer as well as a stem cell marker. 3) Figure 10, Panel A and panel B has the identical heatmap for both males and females. It is not clear why the authors have shown two panels for the same cell lines as cell lines have originated from either males or female derived CRC tissues. Please modify the figure by identifying the gender source of all cell lines and then presenting the heatmap. 4) the study shows some cytotoxicity data from treatment of RV59 to colon cancer cells in vitro.The details on how they derived is not included in the methods.
Reviewer 2 Report
Authors: N. Mokgautsi, T.-H. Huang, Y.-J. Huang et al. Title: Comprehensive Omics Analysis a Novel Small-Molecule Inhibitor of Chemoresistant Oncogenic Signatures in Colorectal Cancer Cell with Antitumor Effects Reviewer's comments: The authors have performed an interesting study in the field of colorectal cancer; their manuscript is well written and designed. I have only minor remarks: 1) There are errors in the Title and Abstract: probably, 'of' is omitted after the word 'Analysis'. In the Abstract, 'epithelial-to-mesenchymal' should be instead of endothelial-to-mesenchymal. In several places of the text, too large empty spaces have been made in the beginings of paragraphs. 2) The authors mention the significance of CSCs and EMT in the pathogenesis of colorectal cancer. However, after reading their manuscript, it seems unclear whether RV59 affects CSCs or EMT.Author Response
Please see the attachment.
